# Risk of Cerebrovascular Events in Deep Brain Stimulation for Parkinson’s Disease Focused on STN and GPi: Systematic Review and Meta-Analysis

**DOI:** 10.3390/brainsci15040413

**Published:** 2025-04-18

**Authors:** Cristofer Zarate-Calderon, Carlos Castillo-Rangel, Iraís Viveros-Martínez, Estefanía Castro-Castro, Luis I. García, Gerardo Marín

**Affiliations:** 1Institute of Brain Research, Universidad Veracruzana, Xalapa 91190, Mexico; cristoferjzc@gmail.com (C.Z.-C.); irais6013@gmail.com (I.V.-M.); luisgarcia@uv.mx (L.I.G.); 2Department of Neurosurgery, “Hospital Regional 1° de Octubre”, Instituto de Seguridad y Servicios Sociales de los Trabajadores del Estado, Mexico City 07300, Mexico; neuro_cast27@yahoo.com; 3Faculty of Medicine, Anáhuac Veracruz—Campus Xalapa, Xalapa 91098, Mexico; estefania.67@hotmail.com; 4Neural Dynamics and Modulation Lab, Cleveland Clinic, Cleveland, OH 44196, USA

**Keywords:** cerebrovascular events, deep brain stimulation, internal globus pallidus, Parkinson’s disease, risk, stroke, subthalamic nucleus

## Abstract

**Background/Introduction**: Parkinson’s disease (PD) is a progressive neurodegenerative disorder treated with deep brain stimulation (DBS) for advanced stages, targeting the subthalamic nucleus (STN) or the internal globus pallidus (GPi). Despite DBS’s symptomatic benefits, cerebrovascular events (CVEs) remain a concern. This study assessed CVE risk in PD patients undergoing DBS. **Methods**: We performed a systematic review and meta-analysis following PRISMA 2020 guidelines. Studies published between 2014 and 2024 that reported CVEs in PD patients treated with DBS-STN or DBS-GPi were included. Data on CVEs, DBS targets, perioperative period, and microelectrode recording (MER) use were extracted, and probability proportions were pooled using a random-effects model. **Results**: Twenty-three studies (4795 patients) were included. The overall CVE probability was 2.71% (95% CI: 2.27–3.18%). Descriptive probabilities were 2.56% (95% CI: 1.94–3.24%) for STN and 0.93% (95% CI: 0.00–3.08%) for GPi. Hemorrhagic events were most common (STN: 2.47%; GPi: 1.98%), while ischemic events were rare (STN: 0.07%; GPi: 1.98%). Note that GPi estimates are based on a considerably smaller population and should be interpreted with caution. Postoperative CVEs (1.74%) were more frequent than intraoperative events (0.17%), and MER use did not significantly alter risk (MER: 2.89% vs. NON-MER: 2.92%). **Conclusions**: Our results suggest that DBS in PD is associated with a relatively low CVE risk (~2.7%), with hemorrhage being the most frequent type; CVEs remain a potential risk factor. Comprehensive evaluation of patient-specific factors and further prospective studies focusing on CVE outcomes are essential to optimize DBS safety in managing PD.

## 1. Introduction

Parkinson’s disease (PD) is a neurodegenerative disease characterized by motor symptoms that include bradykinesia, tremors, rigidity, and balance disturbances [1]. From a neurophysiological perspective, the motor symptoms of PD are often a consequence of abnormalities present in the thalamocortical circuits of the basal ganglia [2]. For this reason, stimulation of these circuits has been proposed as therapeutic targets through deep brain stimulation (DBS). This is a therapeutic option approved by the FDA and which is promising for patients with PD, as it allows for the modulation of brain circuits and human behavior [3]. The benefits of DBS are intrinsically related to the neural response elicited by stimulating cell bodies, nerve terminals, and axons that traverse the stimulation areas [4,5].

The selection of DBS targets in PD is primarily guided by the patient’s predominant symptomatology and the fact that DBS is generally reserved for advanced PD [4]. DBS in PD focuses on two types of stimulation: stimulation of the subthalamic nucleus (STN), which allows for the reduction of medication doses and improvement of motor symptoms, and stimulation of the internal globus pallidus (GPi), where the goal is for the patient to maintain or even increase the medication dose without the risk of dopa-induced dyskinesias. It has been observed that elderly patients tolerate the latter procedure well and that it has an excellent effect on their dyskinesias [6]. The selection between STN and GPi is center and patient, specifically reflecting local expertise and individual clinical profiles. Both methods improve patients’ motor function [7].

However, this therapy involves a surgical implantation procedure that may have adverse effects on patients, such as hemorrhages, infections, device-related issues, and epileptic seizures [8,9]. In this context, hemorrhage is one of the most feared complications due to the neurological deficits it can cause [10]. The estimated risk of symptomatic intracranial hemorrhage ranges from 0.2% to 5.6%, although asymptomatic hemorrhages should not be ruled out [11].

Microelectrode recording (MER) is frequently employed to minimize the risk of suboptimal electrode placement. MER is an intraoperative neurophysiological technique that accurately localizes target brain structures during DBS implantation by providing feedback on real-time neuronal activity. This heightened precision in electrode placement helps reduce complications such as hemorrhage and inadequate stimulation. Multiple studies have shown that MER can improve clinical outcomes by ensuring DBS leads are positioned precisely within the intended target [12,13].

Nevertheless, identifying risk factors contributing to adverse events remains a critical clinical task to mitigate these risks further [14,15].

Given the inherent risks associated with the surgical nature of DBS, it is critical to accurately determine the prevalence and potential hazards of cerebrovascular events (CVEs) in patients with PD. Establishing the likelihood of different types of CVE and understanding their distinct profiles can significantly inform preoperative evaluation, surgical planning, and postoperative care. Moreover, identifying variations in CVE risk between the primary DBS targets (STN and GPi) is essential for optimizing patient selection and tailoring individualized treatment strategies. In this context, the primary objective of this study was to assess the probability of developing CVEs in patients undergoing DBS by examining whether there is an elevated risk depending on the treatment targets, CVE subtype, perioperative period of the event, or the use of MER techniques. Our systematic review and meta-analysis, conducted strictly by PRISMA 2020 guidelines, aims to offer a comprehensive and detailed perspective on these cerebrovascular risks in PD patients receiving DBS, thereby providing valuable insights for clinical practice and future research.

## 2. Materials and Methods

### 2.1. Search Strategy

The present systematic review and meta-analysis followed the 2020 guidelines established by the Preferred Reporting Items for Systematic Reviews and Meta-Analyses (PRISMA) [16]. The study protocol was not registered in PROSPERO or any other database. A comprehensive search was performed across five databases, PubMed, Cochrane Library, Scopus, SciELO, and Google Scholar, to retrieve literature published between 2014 and 2024 in English and Spanish. The search adhered to the PICO principle and utilized free-text phrases tailored to identify studies reporting CVE in PD patients undergoing DBS targeting the STN or GPi. Appendix A provides a detailed summary of the Boolean search strategies and parameters used. Initially, the search yielded 1843 articles before applying time filters and other selection criteria (Figure 1).

### 2.2. Inclusion and Exclusion Criteria

Inclusion: Studies were included if they involved patients diagnosed with PD who underwent DBS that targeted the STN or GPi, reported at least one CVE (stroke, hemorrhage, ischemia, or thrombosis) in this patient population, and included a minimum of four patients receiving DBS in the targeted structures.

Exclusion: Studies were excluded if they were published outside the 2014–2024 period, were conducted in non-human subjects or published in languages other than English or Spanish, or if they were review articles, meta-analyses, case reports, or other non-original research publications. Additionally, studies were excluded if they did not specify the total number of PD patients implanted in STN or GPi, report other pathologies or target structures without providing individual patient numbers or if they did not report any CVE outcomes.

### 2.3. Selection of Studies

In selecting literature, duplicate records were removed using Mendeley Reference Manager (v2.128.0). Two independent reviewers screened titles and abstracts for relevance and classified each study as included or excluded. Discrepancies were resolved by consensus or, if necessary, by a third reviewer.

### 2.4. Data Extraction

Two reviewers independently extracted data on study details (author, publication year, study design, sample size), baseline patient characteristics (number of PD patients who underwent DBS, age, disease specifics, and the country from which data were collected), as well as information related to cerebrovascular outcomes (event type, perioperative period, target structure, and implantation technique: MER or NON-MER). Any discrepancies were resolved through discussion or consultation with a third reviewer.

### 2.5. Quality Assessment

One independent reviewer assessed the quality of the studies included. The Newcastle–Ottawa Scale (NOS) [17] was used for retrospective and prospective cohort studies, while the Cochrane Risk of Bias tool, version 2 (RoB 2) [18] was applied for randomized clinical trials (RCTs). Any disagreements in quality assessments were resolved through collaborative discussion or, if necessary, consultation with a second reviewer.

### 2.6. Statistical Analysis

Meta-analyses were conducted to address the research objectives, employing a significance level of *p* < 0.05 for inferential testing.

#### 2.6.1. Primary Meta-Analysis Model

A random-effects model (using the inverse variance method) was adopted as the primary approach to pool probability proportions of CVEs, accommodating anticipated heterogeneity across studies. Proportions were transformed with the Freeman–Tukey double arcsine method to stabilize variance, particularly in studies with zero events. Between-study variance (τ^2^) was estimated via restricted maximum likelihood (REML). Pooled estimates and 95% confidence intervals (CI) were calculated under both random-effects (with the Hartung–Knapp–Sidik–Jonkman adjustment for conservative inference) and fixed-effects models. Additionally, 95% prediction intervals were computed for the random-effects model. For subgroup analyses with a critically low number of events (typically <10 events), a descriptive approach was employed: crude pooled proportions (total events/total patients at risk) were calculated, with exact 95% CIs derived using the Clopper–Pearson method to reflect the uncertainty associated with rare events transparently.

#### 2.6.2. Overall and Structure-Specific CVE Probability

The primary random-effects model was used independently for studies reporting STN-specific data and for those reporting GPi-specific data. Initially, the primary random-effects model was applied to estimate the probability of CVEs across all eligible DBS-STN and DBS-GPi studies.

Two meta-analyses, one for each structure, were performed to evaluate the structures’ impact on DBS (STN vs. GPi), considering STN and GPi data.

#### 2.6.3. Target-Specific Probability of CVE Subtypes

Separate random-effects meta-analyses were planned to assess the probability of each CVE subtype within the STN and GPi groups.

#### 2.6.4. Perioperative Period and Technique Effects on CVE

Further analyses explored procedure-related factors:Perioperative period: Separate random-effects meta-analyses were conducted to estimate the probability of intraoperative (IO) and postoperative (PO) CVEs, using the total number of DBS patients per study as the denominator. Note that variability in the definition of the PO period may contribute to heterogeneity in these estimates.The technique (MER vs. NON-MER): The influence of MER was evaluated by conducting separate meta-analyses for the MER and NON-MER subgroups using the primary random-effects model.

Forest plots were planned to visualize the results from these analyses.

#### 2.6.5. Heterogeneity and Publication Bias Assessment

Heterogeneity was assessed using the Q statistic, I^2^, and τ^2^ across all meta-analyses. For analyses incorporating 10 or more studies, publication bias was evaluated through funnel plot inspection and, when appropriate, Egger’s regression test. Findings were interpreted cautiously, and the potential impacts of study-level risk of bias and publication bias were considered in the overall synthesis.

All statistical analyses were performed in R (Version 4.4.3) within the RStudio environment for Windows (Version 2024.12.1+563), the *meta* package (Version 8.0-2) was used for conducting meta-analyses.

## 3. Results

### 3.1. Search Results

As depicted in Figure 1, the literature search yielded 23 articles containing data suitable for inclusion in the meta-analysis [9,10,11,19,20,21,22,23,24,25,26,27,28,29,30,31,32,33,34,35,36,37,38]. These articles were carefully evaluated individually against predefined inclusion and exclusion criteria, leading to the selection of 23 studies for meta-analysis (Appendix A).

### 3.2. Patient Characteristics

Data from 23 studies (Table 1) were considered to determine the probability of developing a CVE in PD patients implanted with DBS in the STN or GPi. Of these, 21 were retrospective, 1 was prospective, and 1 was RCT. The overall analysis included 4795 PD patients with DBS, among whom 150 developed either hemorrhagic or ischemic events.

For the structure-specific and CVE subtype analyses, 16 studies were utilized. In the STN group, 2788 patients were included, with 88 CVEs reported (86 hemorrhagic and 2 ischemic), whereas in the GPi group, 6 studies provided data on 252 patients, with 10 CVEs observed (5 hemorrhagic and 5 ischemic).

Regarding the perioperative period and MER analyses, 18 and 17 studies were included, respectively. Specifically, 17 IO and 106 PO CVEs were recorded among 4103 PD-DBS patients. Furthermore, for the localization technique analysis, 15 studies involving patients who underwent DBS with MER comprised 2249 patients and 75 CVEs. In comparison, three studies involving patients implanted without MER (NON-MER) represented a population of 591, with 18 CVEs reported.

Laterality and the number of electrodes implanted in the DBS were not considered for the variables. Similarly, in the PO period, the exact timing of the event was not accounted for, nor was the number of trajectories used in MER. This omission was because most studies did not adequately report these variables in patients with CVEs. Nevertheless, a heterogeneous PO period was observed, ranging from immediately to days, and in the MER data, both single-track and multiple-track approaches were noted.

Other study variables such as age, sex, comorbidities, and ethnicity were excluded from further analysis because over half of the studies did not adequately specify these characteristics for patients who experienced a CVE. Nevertheless, it was observed that the median age of PD patients undergoing DBS was 60.4 ± 10.8 years and that most patients were from European and Asian regions.

The final database can be found in Appendix A.

### 3.3. CVE in PD-DBS Patients

The pooled results of the meta-analysis revealed that the probability of experiencing a CVE in PD patients undergoing DBS was 2.71% (95% CI: 2.27–3.18%), calculated using a random-effects model. The prediction interval ranged from 2.20% to 3.27%. Inter-study heterogeneity was minimal, with an I^2^ = 0, τ^2^ = 0, and a non-significant Q statistic (Q = 16.10, *p* = 0.8107), suggesting consistency across the included studies. The forest plot (Figure 2) illustrates individual study proportions and the combined estimate. While visual inspection of the funnel plot suggested symmetry (Appendix A), the formal Egger’s test for funnel plot asymmetry indicated statistically significant asymmetry (t = 2.13, df = 21, *p* = 0.0456; bias estimate = 0.7820, SE = 0.3679). This result suggests potential publication bias or other small-study effects, warranting caution when interpreting the overall pooled estimate despite the low observed heterogeneity.

### 3.4. CVE per Structure in PD-DBS Patients

For these probabilities, separate meta-analyses were conducted for the STN and GPi structures to estimate the probability of developing a CVE following DBS in each specific structure.

For the STN, under the random-effects model, the meta-analysis yielded a pooled CVE probability of approximately 2.56% (95% CI: 1.94–3.24%). The prediction interval ranged from 1.88% to 3.32%. Heterogeneity across the STN studies was negligible, with I^2^ = 0%, τ^2^ = 0, and a non-significant Q statistic (Q = 12.26, *p* = 0.6594). The forest plot (Figure 3) summarizes the individual study and pooled estimates. Although a visual inspection of the funnel plot (see Appendix A) suggested that studies were primarily within the funnel boundaries, a slight asymmetry was noted. Consistent with this visual observation, Egger’s regression test indicated statistically significant asymmetry (t = 2.22, df = 14, *p* = 0.0438), providing a bias estimate of 0.8896 (SE = 0.4014). This finding potentially suggests the presence of publication bias or small-study effects within this subgroup of studies.

For the GPi, the meta-analysis estimated a pooled CVE probability of 0.93% (95% CI: 0.00–3.08%) under the random-effects model. The prediction interval ranged from 0.00% to 4.56%. Heterogeneity among GPi studies was minimal (I^2^ = 0.0%, τ^2^ = 0; Q = 2.13, df = 5, *p* = 0.8307). The forest plot (Figure 4) displays the individual study estimates and the pooled result. Due to the limited number of included studies (k = 6), Egger’s test was not formally performed; potential publication bias was assessed visually using the funnel plot (see Appendix A).

Given the few studies and patients in GPi, these figures must be interpreted cautiously, and no formal comparison was undertaken.

### 3.5. Type of CVE per Structure

Regarding the probability of developing a specific type of CVE per structure, the results show the following:

#### 3.5.1. Hemorrhages

For STN, the meta-analysis yielded a pooled hemorrhage probability of approximately 2.47% (95% CI: 1.82–3.18%) under the random-effects model. The prediction interval ranged from 1.80% to 3.22%. Heterogeneity was negligible, with I^2^ = 0, τ^2^ = 0, and a non-significant Q statistic (Q = 13.65, *p* = 0.5521). The forest plot (Figure 5) visually summarizes the individual study estimates. In contrast, the funnel plot (see Appendix A) revealed only slight asymmetry, as evidenced by Egger’s test (t = 1.58, df = 14, *p* = 0.1372; bias estimate = 0.7154, SE = 0.4537).

For GPi, due to these limited CVEs (k = 9), a descriptive crude pooled proportion was calculated by summing events and patients, resulting in 1.98%. The corresponding exact 95% confidence interval (Clopper–Pearson method) was [0.65–4.57%].

These results should be interpreted cautiously, as the low number of events and the wide confidence interval underscore the considerable uncertainty in quantifying this specific risk for the GPi target.

#### 3.5.2. Ischemias

Due to the extreme rarity of ischemic events, a standard meta-analysis was not feasible. Instead, a descriptive approach was employed by calculating crude pooled proportions with exact 95% confidence intervals (Clopper–Pearson method).

For STN, 2788 at-risk patients reported two ischemic events, resulting in a crude combined proportion of 0.07% (95% CI: 0.01–0.26%).For GPi, 252 patients at risk reported five ischemic events, yielding a crude combined proportion of 1.98% (95% CI: 0.65–4.57%).

These descriptive estimates for ischemic events should be interpreted cautiously due to the very low number of events observed, particularly for STN, and the consequently wide confidence intervals, which underscore the uncertainty surrounding the precise risk quantification.

### 3.6. Probability of CVE in the Perioperative Period

In terms of CVE probability within a specific perioperative period, the findings indicate the following:

Intraoperative period: For the IO period, the random-effects meta-analysis yielded a pooled CVE probability of about 0.17% (95% CI: 0.00–0.74%), with a prediction interval spanning 0.00% to 3.08%. In contrast to the postoperative findings, heterogeneity was higher (I^2^ = 64.1%, τ^2^ = 0.0026) and the Q statistic was significant (Q = 47.31, *p* = 0.0001). The forest plot (Figure 6) illustrates the individual study and pooled estimates. Inspection of the funnel plot (see Appendix A) indicated a more pronounced asymmetry, which was confirmed by Egger’s regression test (t = 3.89, df = 16, *p* = 0.0013; bias estimate = 2.4730, SE = 0.6361). These results suggest the potential presence of publication bias or small-study effects in the IO period.

Postoperative period: Under the random-effects model, the pooled PO CVEs probability was estimated at approximately 1.74% (95% CI: 0.85–2.86%), with a prediction interval ranging from 0.00% to 5.88%. Heterogeneity was moderate (I^2^ = 56.8%, τ^2^ = 0.0022), and the Q statistic indicated significant variation across studies (Q = 39.40, *p* = 0.0016). The forest plot (Figure 7) summarizes the individual study and pooled estimates. Visual inspection of the funnel plot (see Appendix A) revealed a relatively balanced distribution of studies around the combined estimate. Consistent with this observation, Egger’s test was not significant (t = −0.94, df = 16, *p* = 0.3599; bias estimate = −0.7427, SE = 0.7879), indicating no firm evidence of publication bias or small-study effects in the PO period.

### 3.7. CVE per Technique in PD-DBS Patients

Regarding the probability of developing CVEs based on the use of MER, the results show the following:

MER subgroup: Under the random-effects model, the meta-analysis for the MER subgroup yielded a pooled CVE probability of approximately 2.89% (95% CI: 2.12–3.77%). The prediction interval ranged from 1.81% to 4.18%. Heterogeneity was negligible, with I^2^ = 0%, τ^2^ = 0.0001, and a non-significant Q statistic (Q = 13.17, *p* = 0.5132). The forest plot (Figure 8) summarizes the individual study estimates and the pooled result. While a visual examination of the funnel plot (see Appendix A) showed no pronounced asymmetry, Egger’s test approached but did not reach statistical significance (t = 1.71, df = 13, *p* = 0.1113; bias estimate = 1.0076, SE = 0.5897), suggesting no definitive evidence of publication bias or small-study effects for the MER subgroup.

NON-MER subgroup: For the NON-MER subgroup, the pooled CVE probability was estimated at around 2.92% (95% CI: 0.91–5.87%) under the random-effects model, with a prediction interval of 0.55% to 6.78%. Heterogeneity was minimal (I^2^ = 0%, τ^2^ = 0), and the Q statistic was also non-significant (Q = 1.26, *p* = 0.5326). The forest plot (Figure 9) displays the individual study estimates and the pooled result. Given the limited number of studies (k = 3), Egger’s test was not performed for the NON-MER, but a funnel plot was evaluated for descriptive purposes (see Appendix A).

## 4. Discussion

PD is the second most common neurodegenerative disorder, affecting approximately 1.51 per 1000 individuals worldwide, and its prevalence has significantly increased over the last two decades [39]. Although levodopa provides significant initial relief, its efficacy decreases over time, necessitating advanced interventions such as DBS. DBS involves the implantation of electrodes in specific brain regions (primarily the STN and GPi) to alleviate motor symptoms [40].

Despite its effectiveness, DBS remains an invasive neurosurgical procedure with inherent risks, particularly CVEs. Therefore, this systematic review and meta-analysis aimed to determine the probability of CVEs in PD patients undergoing DBS in the STN or the GPi.

The study explored potential differences in risk according to the target structure, the type of event (hemorrhage or ischemia), the perioperative timing, and the use of MER.

The analysis, which included data from 4795 patients extracted from 23 studies, revealed an overall CVE probability of 2.71% [95% CI: 2.27–3.18%]. This value is in line with previously reported rates for symptomatic intracranial hemorrhages associated with DBS, which range between 0.2% and 5.6% [11,41,42]. This confirms that, although DBS is generally a safe procedure, CVEs constitute a relevant, albeit infrequent, complication [10,43].

However, Egger’s test, with a statistically significant result (*p* = 0.0456), points to a possible publication bias or minor study effects. This suggests that the actual risk might differ slightly and warrants caution when interpreting this estimate.

When analyzing the probability of CVEs by target structure, rates of 2.56% [1.94–3.24%] for DBS-STN and 0.93% [0.00–3.08%] for DBS-GPi were found. It is worth noting that the broad and overlapping CI, particularly for the GPi estimation, derives from a much smaller group of patients (*n* = 252 vs. 2788 for STN), which prevents drawing a statistically significant conclusion regarding the CVEs risk between the structures.

The STN is embedded in a complex vascular environment, irrigated by branches of the posterior cerebral, posterior communicating, and anterior choroidal arteries [44,45,46]. Its proximity to critical perforating vessels and white matter tracts, such as the internal capsule, could theoretically increase its vulnerability during electrode insertion [47,48,49]. In contrast, the GPi, a larger structure with a more lateral location in the basal ganglia, is mainly supplied by the anterior choroidal artery and lenticulostriate branches of the middle cerebral artery [50,51,52,53,54]. Although its size might facilitate trajectory planning, the approaches must traverse zones rich in the perforating arteries essential for the functionality of the basal ganglia [50,55].

Current surgical techniques incorporating high-resolution imaging for trajectory planning and meticulous intraoperative execution help mitigate the inherent anatomical risks associated with DBS. The decision regarding which target to use for DBS is multifactorial. It is influenced by patient age, predominant symptoms (such as motor issues versus dyskinesia), medication-related goals, potential adverse effects, and the surgical team’s expertise [6,7,53,56]. For example, STN DBS is generally associated with reduced levodopa requirements, while GPi DBS may be chosen to address dyskinesia with a lower impact on cognitive or mood functions [40,55,57,58,59]. Our descriptive findings indicate that vascular safety is only one of several important factors in DBS surgery, underscoring the necessity of a tailored treatment approach. Although we report estimates for STN and GPi, it is important to note that the GPi data are derived from a much smaller patient sample; consequently, these estimates should be interpreted cautiously.

On the other hand, when examining the specific CVE, hemorrhagic events predominated, with a probability of 2.47% [1.82–3.18%] in the STN and a descriptive estimate of 1.98% [0.65–4.57%] in the GPi. This pattern is consistent with the proposed mechanism of direct mechanical injury to small vessels during the electrode trajectory [10]. Although not statistically significant, the slightly higher rate of STN could be related to its dense vascular network and proximity to sensitive structures, such as the internal capsule or the choroidal vessels, where minor deviations could impact vessels [41,44,46].

In contrast, although considerably rarer, ischemic events showed a potentially relevant difference: 0.07% [0.01–0.26%] for STN, and 1.98% [0.65–4.57%] for GPi. These estimates, especially in the GPi, are based on a small number of cases and present wide CIs, which limits firm conclusions. However, this trend suggests possible implicated mechanisms. Ischemia in the GPi could result from injury or compression of perforating arteries (such as the lenticulostriate arteries), vasospasm from electrode manipulation, microemboli, or edema affecting perfusion [58,60]. Moreover, the more extensive area that must be traversed to reach the GPi could theoretically increase exposure to these vulnerable small vessels. Although preliminary, this finding suggests that, while hemorrhage is the primary concern in both targets, ischemic mechanisms deserve specific consideration, particularly in procedures directed at the GPi.

Regarding the perioperative period of the CVEs, we found a higher probability in the PO period (1.74% [0.85–2.86%]) compared with the IO period (0.17% [0.00–0.74%]). The IO events probably represent acute hemorrhages detected during or immediately after electrode insertion. The higher PO rate underlines the risk of delayed complications, such as rebleeding from microtrauma, hematoma expansion, consequences of PO hypertension, or effects related to the management of anticoagulant/antiplatelet therapy [9,11,61]. The moderate heterogeneity observed in the PO analysis (I^2^ = 56.8%) could reflect variations in how the PO period was defined among studies, ranging from hours to months. However, the significant heterogeneity and publication bias detected for the IO events (Egger *p* = 0.0013) raises concerns that studies reporting higher IO complication rates may be underrepresented, potentially underestimating the immediate procedural risk.

Encouragingly, the use of MER did not significantly alter the probability of CVEs (MER: 2.89% [2.12–3.77%] vs. NON-MER: 2.92% [0.91–5.87%]). MER is invaluable for physiologically confirming the target, allowing trajectory adjustments to optimize placement and avoid eloquent or vascular structures [12,13]. Our findings suggest that any theoretical increase in risk due to multiple passes is likely offset by the gains in precision and safety offered by real-time feedback. However, the analysis without MER was based on only three studies, limiting this comparison’s robustness.

Although our meta-analysis could not perform detailed subgroup analyses due to limited information in the primary studies, patient factors are undoubtedly crucial. The median age of patients in the included studies was approximately 60.4 years, which, while reflecting the typical candidates for DBS, does not account for the fact that many patients are older, frequently exceeding 65–70 years. Advanced age is associated with increased vascular fragility and a higher prevalence of comorbidities such as hypertension, diabetes, and atherosclerosis, all of which are independent risk factors for CVEs [62,63,64,65,66]. Hypertension, in particular, is common in the DBS population for Parkinson’s (potentially >60% [61,66,67]) and requires meticulous perioperative control. Moreover, although most patients in our included studies came from European or Asian centers, potential differences in baseline vascular health or surgical practices related to ethnicity or geographic region may exist and were not assessable [65,68,69].

These unmeasured factors contribute to the complexity of individualized risk assessment and underscore the need for comprehensive baseline evaluations and optimization of medical or other comorbidities that may arise before DBS surgery [61,70].

In summary, this meta-analysis demonstrates a low (~2.7%) but clinically relevant risk of CVEs associated with DBS for PD. Although hemorrhage is the predominant subtype, the rare ischemic events warrant specific attention, particularly in trajectories toward the GPi. Moreover, the current aggregated evidence suggests that neither the choice of structure (STN or GPi) nor the use of MER significantly alters the CVE risk, indicating that surgical expertise and preoperative planning are fundamental. The elevated risk in the PO period highlights the need for vigilant monitoring and aggressive management of contributing factors such as blood pressure fluctuations. Furthermore, the preoperative management of medications, such as carefully discontinuing anticoagulants or antiplatelet agents and optimizing antihypertensive therapy, is crucial in minimizing CVE risk [41,71,72,73].

These findings reinforce the necessity of individualized patient selection and careful surgical planning. Neurosurgeons and multidisciplinary teams must weigh specific therapeutic objectives against the patient’s vascular profile, the anatomical challenges of the chosen target, and the potential for perioperative complications. The acquisition of advanced preoperative imaging to visualize individual vascular anatomy concerning planned trajectories could play an increasingly important role in risk minimization [63,74,75]. Likewise, the cumulative surgical experience, reflected in the learning curve of DBS procedures, has been associated with a reduction in complication rates, including CVE, which underscores the importance of center volume and surgeon proficiency [19,76].

Looking to the future, large-scale, multicenter prospective registries with standardized data collection, including detailed patient characteristics (such as comorbidities and ethnicity), precise surgical techniques (laterality, MER details, trajectories), specific definitions of CVEs, and long-term follow-up, are needed to refine our understanding of CVE risks, more clearly identify modifiable risk factors, and further improve the safety of this transformative therapy for patients with PD [63,75,77,78].

## 5. Limitations

The study acknowledges that, because it predominantly relies on retrospective investigations, there is an inherent risk of bias in data selection and reporting, which could affect the reliability of the results. Additionally, the inclusion criterion that required reporting at least one CVE event may have skewed the prevalence toward higher values by excluding studies that did not report any event. This situation, along with the low frequency of ischemic events and the limited availability of specific data (such as sex, comorbidities, and surgical details), prevented subgroup analyses that could have identified particular risk factors and limited the temporal scope to perioperative events without exploring long-term outcomes.

On the other hand, a notable imbalance is observed between study targets, as there is a marked disparity between the patient populations with PD in STN when compared with GPi. These differences reduce the statistical power of GPi-specific analyses, generating wide confidence intervals and complicating direct comparisons between the two targets. Moreover, the asymmetry in publication bias analyses and the influence of smaller studies suggest possible publication or small-study effects, underscoring the need for future large-scale, prospective studies with standardized data collection.

## 6. Conclusions

In conclusion, the overall probability of CVE in PD patients undergoing DBS was 2.71% (95% CI: 2.27–3.18%). For STN, the pooled probability was 2.56% (95% CI: 1.94–3.24%), and for GPi, it was 0.93% (95% CI: 0.00–3.08%). Given the wider confidence intervals and smaller sample sizes for GPi, these estimates should be interpreted cautiously. Hemorrhagic events were the most common, observed in 2.47% (95% CI: 1.82–3.18%) of STN cases and 1.98% (95% CI: 0.65–4.57%) of GPi cases. Ischemic events were rare, with estimates of 0.07% (95% CI: 0.01–0.26%) in STN and 1.98% (95% CI: 0.65–4.57%) in GPi; however, the low number of events limits the precision of these estimates. Regarding other factors, such as the perioperative period, CVEs were more frequent in the PO (1.74%, 95% CI: 0.85–2.86%) than in the IO (0.17%, 95% CI: 0.00–0.74%), highlighting the importance of PO monitoring. Additionally, the use of MER does not appear to significantly affect the probability of CVEs, with rates of 2.89% (95% CI: 2.12–3.77%) in the MER and 2.92% (95% CI: 0.91–5.87%) in the NON-MER subgroup. This suggests that the precision benefits of MER may compensate for any theoretical increase in risk due to multiple passes. However, the NON-MER analysis was based on only three studies, limiting its robustness.

Overall, these descriptive findings indicate that, although the overall probability of CVE in PD patients undergoing DBS is relatively low, CVEs must still be regarded as a potential risk factor during DBS procedures. Therefore, a thorough evaluation of individual patient factors, such as symptom profile, medication requirements, and vascular risk, is essential in planning DBS surgery, in conjunction with the use of advanced imaging and surgical expertise to enhance safety. Future prospective studies focusing on CVE outcomes are needed to refine these estimates further and optimize the safety and efficacy of DBS in managing PD.

## Figures and Tables

**Figure 1 brainsci-15-00413-f001:**
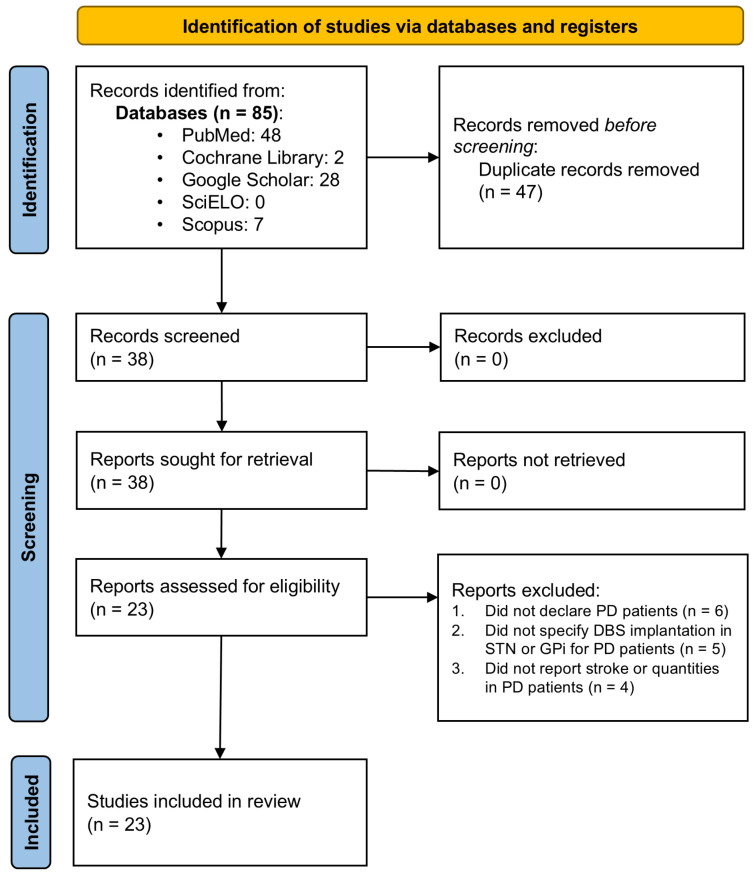
PRISMA flow diagram. Flow chart of literature search strategies.

**Figure 2 brainsci-15-00413-f002:**
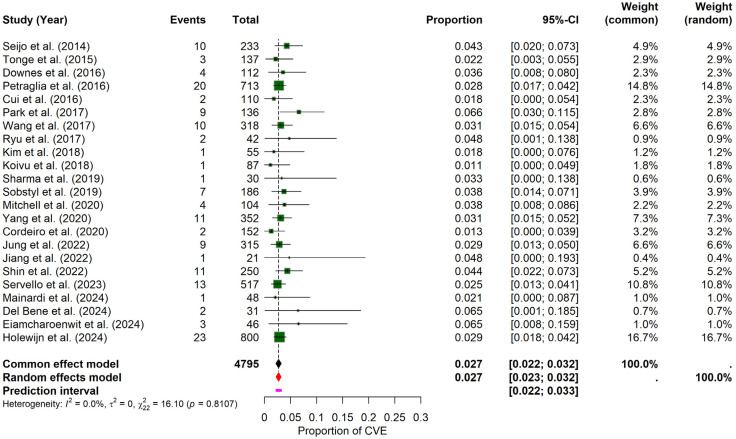
General probability of CVE in PD–DBS. This forest plot shows the overall pooled probability of CVE across all included PD patients undergoing DBS, showing individual estimates and the combined effect [9,10,11,19,20,21,22,23,24,25,26,27,28,29,30,31,32,33,34,35,36,37,38].

**Figure 3 brainsci-15-00413-f003:**
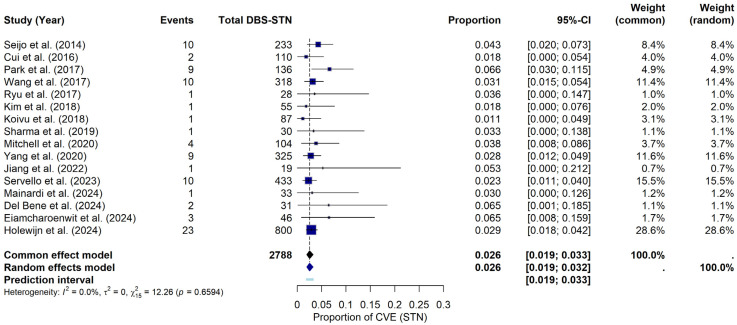
CVE probability in DBS-STN. Forest plot illustrating the pooled probability of CVE, specifically in PD patients with DBS-STN, along with point estimates and confidence intervals for each study [9,19,22,23,24,25,26,27,28,29,32,34,35,36,37,38].

**Figure 4 brainsci-15-00413-f004:**
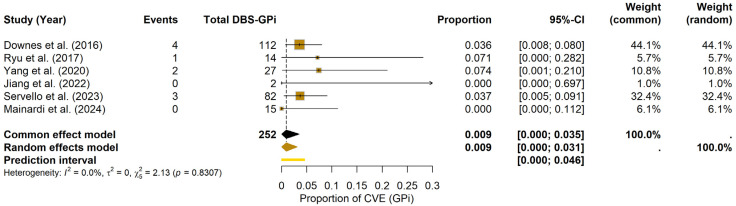
CVE probability in DBS-GPi. Forest plot depicting the pooled probability of CVE in PD patients undergoing DBS-GPi, with individual study results and the summary measure [20,24,29,32,34,35].

**Figure 5 brainsci-15-00413-f005:**
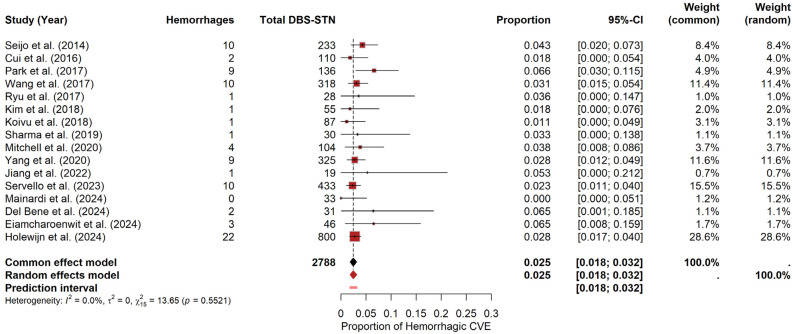
Hemorrhage risk in DBS-STN. Forest plot summarizing hemorrhage probability for PD patients undergoing DBS-STN. Each study’s estimate and confidence interval appear, along with the pooled estimate [9,19,22,23,24,25,26,27,28,29,32,34,35,36,37,38].

**Figure 6 brainsci-15-00413-f006:**
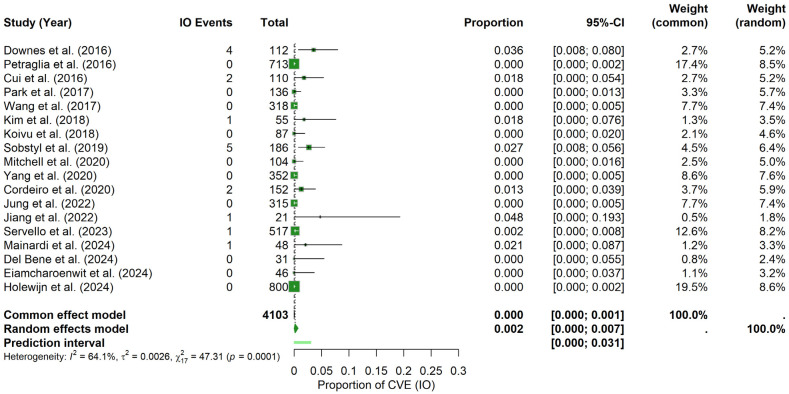
Intraoperative CVE probability. Forest plot shows the probability of CVEs occurring intraoperatively in DBS surgery, combining data from multiple studies into a pooled effect [9,11,20,21,22,23,25,26,28,29,30,31,32,34,35,36,37,38].

**Figure 7 brainsci-15-00413-f007:**
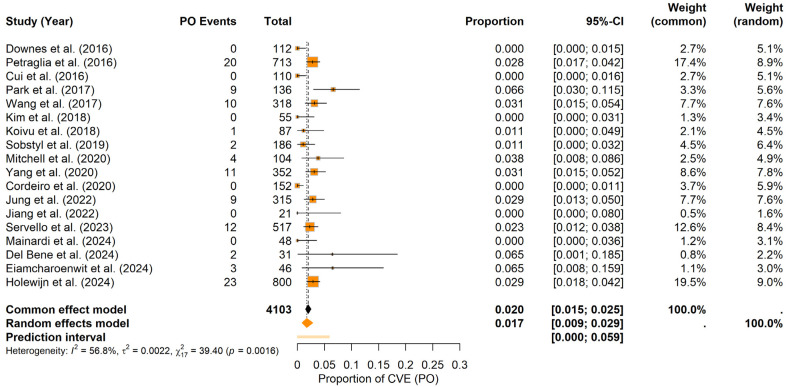
Postoperative CVE probability. Forest plot summarizing the probability of CVEs in the PO period in DBS surgery in PD patients. Individual study estimates and confidence intervals are presented, culminating in a pooled estimate [9,11,20,21,22,23,25,26,28,29,30,31,32,34,35,36,37,38].

**Figure 8 brainsci-15-00413-f008:**
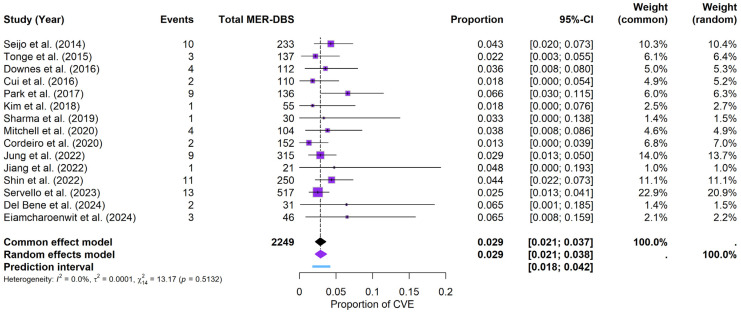
CVE probability with MER. Forest plot focusing on MER in DBS surgery. The figure compares individual estimates, CI, and the pooled probability [9,10,19,20,22,25,27,28,30,31,32,33,34,36,37].

**Figure 9 brainsci-15-00413-f009:**
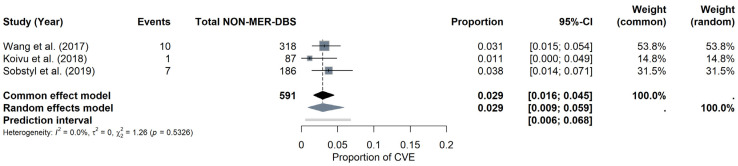
CVE probability without MER (NON-MER). Forest plot highlighting studies that did not employ MER in DBS surgery. Individual study results are illustrated, and a pooled CVE estimate is provided at the bottom [11,23,26].

**Table 1 brainsci-15-00413-t001:** Characteristics of the selected studies. This table summarizes key information from the studies. Study design: the type of study, where R = retrospective, P = prospective, and RCT = randomized controlled trial. Total number of DBS patients: Number of PD patients undergoing DBS. Structure: The DBS structure where the CVE occurred. Type of CVE: Type of reported CVE, where H = hemorrhage and I = ischemia. Number of CVEs: Total number of CVEs recorded. Quality: Quality assessment score, based on the NOS for all studies, except for Del Bene et al., 2024 [36], was evaluated using the RoB 2. Use of MER: Indicates whether MER was used during DBS surgery. Age: Mean or median age of the overall study population (L = left, R = right, Y = younger, O = older, WO = with complications, WOC = without complications, IQR = interquartile range). Country: The country in which the study was conducted.

Author and Year	Study Design	Total Number of DBS Patients	Structure	Type of CVE	Number of CVE	Quality	Use of MER	Age	Country
Seijo et al., 2014 [19]	R	233	STN	H	10	7	YES	61.09 (7.8)	Spain
Tonge et al., 2015 [10]	R	137	-	H	3	7	YES	57.0 (13.6)	Netherlands and Turkey
Downes et al., 2016 [20]	R	112	GPi	I	4	7	YES	60.3 (14.3)	United States
Petraglia et al., 2016 [21]	R	713	-	H	20	9	-	61.1 (9.9)	United States
Cui et al., 2016 [22]	P	110	STN	H	2	6	YES	58 (IQR: 39–77)	China
Park et al., 2017 [9]	R	136	STN	H	9	7	YES	55.14 (13.8)	South Korea
Wang et al., 2017 [23]	R	318	STN	H	10	7	NO	55.7 (14.8)	China
Ryu et al., 2017 [24]	R	42	STN and GPi	H and I	2	8	-	STN: 56.9 (7.7); GPi: 57.9 (8.4)	South Korea
Kim et al., 2018 [25]	R	55	STN	H	1	7	YES	Y: 56.7 (5.7); O: 68.5 (2.9)	South Korea
Koivu et al., 2018 [26]	R	87	STN	H	1	6	NO	61 (IQR: 54–65)	Finland
Sharma et al., 2019 [27]	P	30	STN	H	1	7	YES	77.5 (2.1)	United States
Sobstyl et al., 2019 [11]	R	186	-	H	7	7	NO	-	Poland
Mitchell et al., 2020 [28]	R	104	STN	H	4	9	YES	Y: 60.8 (7.1); O: 77.6 (2.8)	United States
Yang et al., 2020 [29]	R	352	STN and GPi	H	11	7	-	62.22 (6.08)	China
Cordeiro et al., 2020 [30]	R	152	-	H	2	7	YES	64.7 (10.4)	United States
Jung et al., 2022 [31]	R	315	-	H	9	6	YES	57.5 (11.9)	South Korea
Jiang et al., 2022 [32]	R	21	STN and GPi	H	1	7	YES	75 (IQR: 75–85)	China
Shin et al., 2022 [33]	R	250	-	H	11	9	YES	-	South Korea
Servello et al., 2023 [34]	R	517	STN and GPi	H	13	7	YES	61 (IQR: 35–77)	Italy
Mainardi et al., 2024 [35]	R	48	STN and GPi	I	1	9	-	STN: 58 (IQR 14); GPi: 61 (IQR 11.5)	Italy
Del Bene et al., 2024 [36]	RCT	31	STN	H	2	Low Risk	YES	L-STN: 56.7 (8.6); R-STN: 58.9 (6.3)	United States
Eiamcharoenwit et al., 2024 [37]	R	46	STN	H	3	7	YES	WC: 58 (IQR: 53–66); WOC: 59 (IQR: 55–65)	Thailand
Holewijn et al., 2024 [38]	R	800	STN	H and I	23	7	-	61.1 (8.4)	Netherlands

## Data Availability

The list of studies included, and the data used for statistical analyses can be found in Appendix A. Further inquiries can be directed to the corresponding author.

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
