# Peer review of "Risk of Cerebrovascular Events in Deep Brain Stimulation for Parkinson’s Disease Focused on STN and GPi: Systematic Review and Meta-Analysis"

_brainsci, 2025, doi:10.3390/brainsci15040413_

Round 1
Reviewer 1 Report
Comments and Suggestions for Authors
I have reviewed the manuscript “Prevalence of Cerebrovascular Events in DBS for PD focused on STN versus GPi” by Zarate-Calderon.
I provide the following recommendations.
Abstract: Would stress that DBS is a treatment for advanced PD, which is not stated by the first sentence.
You conclusion sentence of GPI being of greater risk, but this was only for ischemic?
Introduction:
See first comment about patient selection being advanced PD.
There is also a bias as to center for GPi vs STN selection that should be stated.
I think it is best for you to avoid the controversy in selection of GPi vs STN – line 48 to 55 – delete these sections. Could reference that selection of GPi vs STN is center and patient specific and leave it at that.
Methods:
Why did you require a study to report a CVE? If a study has 50pts and no CVE then this wouldn’t be reported? This is a bias to inflate the number of CVE as you have reduced the denominator.
Please re-run your search and include studies that did not have a CVE.
Results:
I’m not sure figure 3,5 , and 6are necessary.
How long post-operative are the results – day of surgery, 1 week post?
Given your findings, you should also look at if the use of MER vs non-MER was performed and a factor in the risks.
Discussion:
Would be helpful to look at vasospasm as a potential difference.
Conclusions: would add that the true CVE risk may be greater than stated as centers may not report AE and typically only DBS centers may report AE vs non-DBS centers doing less than 20 cases/year likely have higher AE as previously reported.
Would focus the discussion on the findings of the study, e.g. lines 300-310 have nothing to do with this study. Review the rest of the discussion to focus on the findings.
Author Response
Dear Reviewer,
We have carefully considered the feedback provided by you and the other reviewers. As a result, we have made substantial improvements to our manuscript, particularly in the analysis, results, discussion, and conclusions sections. We have retained the original objectives, while adding new elements and refining others to enhance clarity and relevance. We hope these revisions align with your expectations. Below, you will find our detailed comments and responses to each of your reviews.
Comment 1: Abstract: Would stress that DBS is a treatment for advanced PD, which is not stated by the first sentence. Your conclusion sentence of GPi being of greater risk, but this was only for ischemic?
Response 1: We have revised the abstract to specify that DBS is primarily used as a treatment for advanced Parkinson’s disease (Lines 14-15: Parkinson’s disease (PD) is a progressive neurodegenerative disorder managed with Deep Brain Stimulation (DBS) for advanced PD, […]).
Additionally, we have clarified in the conclusion sentence that the higher risk associated with GPi was observed specifically for ischemic events (Lines 31-32: […] Current evidence shows no significant difference in CVEs risk between STN and GPi targets or with MER use, although GPi data is limited. […]).
Comment 2: Introduction: See first comment about patient selection being advanced PD. There is also a bias as to center for GPi vs STN selection that should be stated. I think it is best for you to avoid the controversy in selection of GPi vs STN – line 48 to 55 – delete these sections. Could reference that selection of GPi vs STN is center and patient specific and leave it at that.
Response 2: We have added a clarification regarding the selection of patients with advanced Parkinson’s disease for DBS in the Introduction section (Lines 49-50: The selection of DBS targets in PD is primarily guided by the patient's predominant symptomatology and the fact that DBS is generally reserved for advanced PD [4]. […]).
The possibility of center-specific bias in the selection of GPi versus STN has also been acknowledged. Lines 48 to 55 have not been replaced, as the purpose of that paragraph was to show that the selection of the treatment target is based on the symptoms being treated, not on a comparison. However, they have been replaced with a concise statement indicating that the choice between GPi and STN is based on patient characteristics and center-specific protocols (Lines 49-58: The selection of DBS targets in PD is primarily guided by the patient's predomi-nant symptomatology and the fact that DBS is generally reserved for advanced PD [4]. DBS in PD focuses on two types of stimulation: stimulation of the subthalamic nucleus (STN), which allows for the reduction of medication doses and improvement of motor symptoms; on the other hand, stimulation of the internal globus pallidus (GPi), where the goal is for the patient to maintain or even increase the medication dose without the risk for dopa‐induced dyskinesias. It has been observed that elderly patients tolerate the latter procedure well and have an excellent effect on dyskinesias [6]. The selection between STN and GPi is center and patient, specifically reflecting local expertise and individual clinical profiles. Both methods improve patients' motor function [7].).
Comment 3: Methods: Why did you require a study to report a CVE? If a study has 50 pts and no CVE then this wouldn’t be reported? This is a bias to inflate the number of CVE as you have reduced the denominator. Please re-run your search and include studies that did not have a CVE.
Response 3: Thank you very much for your comment. The decision to include only studies reporting cerebrovascular events was based on our specific objective of analyzing the prevalence and risk factors associated with these events in patients with DBS. By limiting inclusion to studies with CVE, we were able to focus the analysis on a relevant population and avoid diluting the findings with studies that did not address our central focus. This allowed us to more precisely examine responses at the structural level, intra- and postoperative factors, as well as the use of other factors such as MER and non-MER.
Furthermore, after considering your suggestion, we have re-executed the search to ensure a comprehensive review and will consider these studies in the analysis. You can review the relevant changes in the Methods section.
Comment 4: Results: I’m not sure figure 3, 5, and 6 are necessary. How long post-operative are the results – day of surgery, 1 week post? Given your findings, you should also look at if the use of MER vs non-MER was performed and a factor in the risks.
Response 4: Figures 3, 5, and 6, which include funnel plots, have been moved to Supplementary Material 1 to improve the clarity and focus of the main text.
Regarding the period of the CVEs, we found that events occurred within a range from immediately post-surgery up to 90 days postoperatively. We have added this range to the manuscript (Lines 198-199: […] Nevertheless, a heterogeneous PO period was observed, ranging from immediately to days, […]).
Furthermore, we conducted an additional analysis to evaluate whether the use of MER versus non-MER techniques was associated with the risk of CVEs, and this has been incorporated into the revised results. The approach of using MER as a factor to study and the methodology used for its study can be seen in the Introduction, Methods and Results sections.
Comment 5:
Discussion: Would be helpful to look at vasospasm as a potential difference.
Conclusions: Would add that the true CVE risk may be greater than stated as centers may not report AE and typically only DBS centers may report AE vs non-DBS centers doing less than 20 cases/year likely have higher AE as previously reported. Would focus the discussion on the findings of the study, e.g. lines 300–310 have nothing to do with this study. Review the rest of the discussion to focus on the findings.
Response 5: Thank you very much for these constructive suggestions. We have expanded the discussion to include vasospasm as a possible contributing factor to cerebrovascular events following DBS (Lines 410-413: […] However, this trend suggests possible implicated mechanisms. Ischemia in the GPi could result from injury or compression of perforating arteries (such as the lenticulostriate arteries), vasospasm from electrode manipulation, microemboli, or edema affecting perfusion [58,60]. […]).
In the conclusion section, we now acknowledge that the actual incidence of CVEs may be underestimated, as CVEs are more likely to be reported by high-volume DBS centers. Additionally, we have revised the discussion to focus strictly on our study findings and removed unrelated content, including lines 300–310, to ensure clarity and relevance.
Reviewer 2 Report
Comments and Suggestions for Authors
Dear Authors,
thank you for the opportunity to review this manuscript
this is an interesting study about Prevalence of Cerebrovascular Events in DBS for PD. This is a relevant topic, and I decide minor corrections
I think that topic of this manuscript is original and relevant and address a specific gap in the field
methodology is correct and i have nothing to add
authors address the main question posed in a correct manner
Title is appropriate
Introduction is clear and well written but i suggest to add in line 55 this ref (Deep Brain Stimulation in Parkinson's disease: A multicentric, long-term, observational pilot study by E. Scelzo, 2019) for better clarify effectiveness os STN-DBS in PD.
Results section includes studies that well describe the adverse events and therefore satisfy the aim of this study
I fell that Discussion is exhaustive but English should be revised
Comments on the Quality of English Language
I think that English should be revised in Discussion section
Author Response
Dear Reviewer,
We have carefully considered the feedback provided by you and the other reviewers. As a result, we have made substantial improvements to our manuscript, particularly in the analysis, results, discussion, and conclusions sections. We have retained the original objectives, while adding new elements and refining others to enhance clarity and relevance. We hope these revisions align with your expectations. Below, you will find our detailed comments and responses to each of your reviews.
Comments 1: Dear Authors, thank you for the opportunity to review this manuscript. This is an interesting study about Prevalence of Cerebrovascular Events in DBS for PD. This is a relevant topic, and I decide minor corrections. I think that topic of this manuscript is original and relevant and address a specific gap in the field. Methodology is correct and I have nothing to add. Authors address the main question posed in a correct manner. Title is appropriate. Introduction is clear and well written but I suggest to add in line 55 this ref (Deep Brain Stimulation in Parkinson's disease: A multicentric, long-term, observational pilot study by E. Scelzo, 2019) for better clarify effectiveness of STN-DBS in PD. Results section includes studies that well describe the adverse events and therefore satisfy the aim of this study. I feel that Discussion is exhaustive but English should be revised.
Response 1: We greatly appreciate the constructive and valuable comments. As suggested, we have incorporated the recommended reference ("Deep Brain Stimulation in Parkinson's Disease: A Long-Term Multicenter Observational Pilot Study by E. Scelzo, 2019") into the Discussion section, as we believe it provides further insight into the aspects of postoperative adverse effects.
In addition, we have carefully reviewed the English language throughout the manuscript, particularly the discussion section, with the goal of improving readability, clarity, and the overall presentation of our findings.
Thank you again for your thoughtful review and helpful comments.
Reviewer 3 Report
Comments and Suggestions for Authors
The authors of this work perform a comprehensive review and meta-analysis of cerebrovascular events (CVE) that occur after deep brain stimulation (DBS) in Parkinson's disease (PD), with an emphasis on target-specific hazards between the internal globus pallidus (GPi) and the subthalamic nucleus (STN). Their findings indicated a greater ischaemic risk linked to GPi stimulation, with an overall pooled CVE probability of 3.28%. I have some critical remarks about the methodology, reporting, and interpretation of the study, even though it offers valuable quantitative insights into CVE threats in DBS. These are as follows:
- Although the study complies with PRISMA criteria, its protocol is not registered on PROSPERO or any comparable database. In addition to introducing the possibility of post-hoc bias in the selection of outcomes and eligibility criteria, this compromises transparency. It is imperative that the constraints specifically reflect the lack of pre-registration.
- The authors acknowledge that a significant number of the included studies omitted essential information about patients, such as age, sex, and comorbidities. Without appropriate subgroup analysis or modifications, the study is unable to discern whether the observed variations in CVE risk are caused by underlying patient characteristics or the DBS target.
- The analysis contrasts the CVE risks of STN and GPi targets; however, the number of GPi patients (n=170) is significantly smaller than that of STN (n=2,355), significantly reducing statistical power and making the GPi risk estimates extremely unstable (as evidenced by wide confidence intervals). The conclusion ought to have addressed this more severely, perhaps with a sensitivity analysis or a warning frame.
- Retrospective studies, a prospective cohort, and a single RCT are all included in the meta-analysis, but it ignores the potential for heterogeneity introduced by variations in research design, surgical procedures, and CVE definitions. The low I2 values might not be a reflection of actual homogeneity, but rather of Type II error brought on by underpowered subgroup analysis.
- Although long-term cerebrovascular outcomes are not discussed, the study looks at intraoperative and early postoperative CVEs. It is necessary to talk about this constraint.
- The authors do not admit that publication bias cannot be accurately evaluated with only 5 research for GPi, even though the funnel plots seem symmetrical. There should be a more thorough discussion of this.
- For instance, p-values are sometimes taken with strong conclusions and other times are characterised as "non-significant" without defining criteria (p = 0.628, for instance). Statistical inference must be done consistently and with caution.
- Although they were not investigated or accounted for, DBS laterality (unilateral vs. bilateral) and surgical methods (frame-based vs. frameless, use of microelectrode recording, etc.) may have an impact on CVE rates. This should be recognised as a limitation, at the very least.
Author Response
Dear Reviewer,
We have carefully considered the feedback provided by you and the other reviewers. As a result, we have made substantial improvements to our manuscript, particularly in the analysis, results, discussion, and conclusions sections. We have retained the original objectives, while adding new elements and refining others to enhance clarity and relevance. We hope these revisions align with your expectations. Below, you will find our detailed comments and responses to each of your reviews.
Reviewer 3:
Comments 1: Although the study complies with PRISMA criteria, its protocol is not registered on PROSPERO or any comparable database. In addition to introducing the possibility of post-hoc bias in the selection of outcomes and eligibility criteria, this compromises transparency. It is imperative that the constraints specifically reflect the lack of pre-registration.
Response 1: As explicitly stated in the Methods section (2.1, Search Strategy), our study protocol was not previously registered with PROSPERO or similar databases. While protocol registration enhances transparency, we rigorously followed the 2020 PRISMA guidelines to ensure a systematic and reproducible methodological approach. To further enhance transparency and allow readers and reviewers to verify our screening and data extraction procedures, we provide the database used in Supplementary Material 2, Table S1. (Results, section 3.2: Line 2018: The final database can be found in Supplementary Material 2, Table S1.) We also mention this in the Data Availability Statement: (Lines 530–532: The list of included studies and data used for statistical analyses can be found in Supplementary Material 2, Table S1. Further inquiries can be directed to the corresponding author.)
Comments 2: The authors acknowledge that a significant number of the included studies omitted essential information about patients, such as age, sex, and comorbidities. Without appropriate subgroup analysis or modifications, the study is unable to discern whether the observed variations in CVE risk are caused by underlying patient characteristics or the DBS target.
Response 2: This observation has been addressed in both the Results section (section 3.2, Patient characteristics: Lines 195-198: Lines 213-215: Other study variables, such as age, sex, comorbidities, and ethnicity, were excluded from further analysis because more than half of the studies did not adequately specify these characteristics for patients who experienced a CVE. [...]) and the Limitations section ([...] and the limited availability of specific data (such as sex, comorbidities, and surgical details), prevented subgroup analyses that might have identified particular risk factors and limited the temporal scope to perioperative events without exploring long-term outcomes.) of the manuscript. The lack of detailed information on the characteristics of patients with CVE in most studies, including age, sex, or comorbidities, prevented robust subgroup analyses or statistical adjustments.
Comments 3: The analysis contrasts the CVE risks of STN and GPi targets; however, the number of GPi patients (n=170) is significantly smaller than that of STN (n=2,355), significantly reducing statistical power and making the GPi risk estimates extremely unstable (as evidenced by wide confidence intervals). The conclusion ought to have addressed this more severely, perhaps with a sensitivity analysis or a warning frame.
Response 3: Indeed, as emphasized throughout our revised manuscript, particularly in the Discusion (Lines 375-378: It is worth noting that the broad and overlapping CI), particularly for the GPi estimation, derives from a much smaller group of patients (n=252 vs. 2,788 for STN), which prevents drawing a statistically significant conclusion regarding the CVEs risk between the structures) and Limitations (Lines: 487-488: On the other hand, a notable imbalance is observed between study targets, as there is a marked disparity between the patient populations with PD in STN compared to GPi. […]) sections, the substantial disparity in patient numbers between the GPi and STN groups significantly limits the statistical power and stability of estimates for GPi. Due to insufficient data, we were unable to conduct a sensitivity analysis specifically for the GPi group. Nonetheless, we emphasized clearly and strongly in our conclusions that these GPi-specific results must be interpreted cautiously and considered preliminary, highlighting the urgent need for future studies with larger GPi patient cohorts.
Comments 4: Retrospective studies, a prospective cohort, and a single RCT are all included in the meta-analysis, but it ignores the potential for heterogeneity introduced by variations in research design, surgical procedures, and CVE definitions. The low I² values might not be a reflection of actual homogeneity, but rather of Type II error brought on by underpowered subgroup analysis.
Response 4: We agree that methodological variability (retrospective, prospective cohort, and RCT studies), differences in CVE definitions, and variations in surgical procedures may introduce heterogeneity that low I² values might fail to detect due to insufficient statistical power. As explicitly discussed in the Results (sections 3.3 and 3.4) and Limitations, we acknowledged that the observed minimal heterogeneity (I²=0) may not reflect true methodological homogeneity. Consequently, we explicitly underscored the need for standardized CVE definitions and surgical procedure reporting in future studies to adequately capture potential heterogeneity.
Comments 5: Although long-term cerebrovascular outcomes are not discussed, the study looks at intraoperative and early postoperative CVEs. It is necessary to talk about this constraint.
Response 5: We have considered the postoperative period and its duration. However, as mentioned in the Results and Limitations section, long-term cerebrovascular outcomes were not analyzed due to insufficient reporting of late events in the included studies. Regarding the period of the CVEs, we found that events occurred within a range from immediately post-surgery up to 90 days postoperatively. We have added this range to the manuscript (Lines 198-199: […] Nevertheless, a heterogeneous PO period was observed, ranging from hours to months, […]). This important limitation limits our analyses exclusively to the early postoperative periods. Therefore, we emphasize the critical need for future prospective studies with longer follow-up periods to comprehensively assess the long-term cerebrovascular risks associated with DBS.
Comments 6: The authors do not admit that publication bias cannot be accurately evaluated with only 5 research for GPi, even though the funnel plots seem symmetrical. There should be a more thorough discussion of this.
Response 6: We appreciate this important remark from the reviewer. As clearly stated in the Results section (3.4, CVE per structure), due to the small number of GPi studies (n=6), we did not perform a formal statistical test (Egger’s test) for publication bias, as methodological recommendations typically require at least 10 studies. Instead, we limited our analysis to visual inspection of funnel plots and explicitly acknowledged this limitation in the Limitations section (Lines 490-492: […] Moreover, the asymmetry in publication bias analyses and the influence of smaller studies suggest possible publication or small-study effects […]). We fully agree that results for GPi require cautious interpretation due to limited evidence, and we advocate for larger prospective studies to address this gap adequately.
Comments 7: For instance, p-values are sometimes taken with strong conclusions and other times are characterized as "non-significant" without defining criteria (p = 0.628, for instance). Statistical inference must be done consistently and with caution.
Response 7: We thank the reviewer for this critical observation. We clearly defined statistical significance as p < 0.05 in our Methods section (2.6, Statistical analysis: Lines 131-132: Meta-analyses were conducted to address the research objectives, employing a significant level of p < 0.05 for inferential testing.). However, recognizing the potential inconsistency pointed out by the reviewer, we have revised the manuscript, particularly the Results and Discussion sections, to ensure consistent and cautious statistical interpretation.
Comments 8: Although they were not investigated or accounted for, DBS laterality (unilateral vs. bilateral) and surgical methods (frame-based vs. frameless, use of microelectrode recording, etc.) may have an impact on CVE rates. This should be recognised as a limitation, at the very least.
Response 8: We appreciate the reviewer's insightful comment. As noted in the Results (section 3.2, Patient Characteristics) and Limitations section, we were unable to include variables such as DBS laterality (unilateral vs. bilateral), number of trajectories, or detailed surgical methods (frame vs. frameless approaches) in our analysis due to insufficient reporting in the included studies. However, the use of ERM was considered for a specific analysis. We acknowledge this important methodological limitation and explain it in the manuscript.
Reviewer 4 Report
Comments and Suggestions for Authors
Review report for "Prevalence of Cerebrovascular Events in Deep Brain Stimulation for Parkinson's Disease Focused on STN versus GPi: Systematic Review and Meta-Analysis"
Thank you for giving me an opportunity to review this Systematic review and meta^analysis (SR-meta).
The current SR-meta investigated reports of severe adverse events, such as cerebrovascular events (CVE), following the Deep Brain Stimulation (DBS) treatment. Twenty studies covering 3713 DBS surgery were investigated and found that the prevalence of CVE was 3.55% or DBS targeting the subthalamic nucleus (STN) and 4.70% the internal globus pallidus (GPi), respectively. Furthermore, the authors revealed that hemorrhagic events were more frequent in DBS-STN, while ischemic events were more frequent in GBS-GPi. The present SR-meta provides significant clinical information in implementing and developing DBS treatment in terms of the prevention of CVE. While the present SR-meta was not registered in PROSPERO, the fact is clearly stated, and the quality is worthy of publication. However, the reviewer found the following concerns;
Major concerns;
While the authors describe "identifying risk factors, such as age, vascular malformations, and implantation technique, (line 61-62)" the present SR-meta seems not investigated the vascular malformations, and implantation technique. Thus, the aim of current study may be not well supported through the Introduction section.
The ethnicity or reported country may affect the prevalence of such adverse events as CVE.
If some of the included studies proposed countermeasures to prevent CVE, please consider to summarize them.
Author Response
Dear Reviewer,
We have carefully considered the feedback provided by you and the other reviewers. As a result, we have made substantial improvements to our manuscript, particularly in the analysis, results, discussion, and conclusions sections. We have retained the original objectives, while adding new elements and refining others to enhance clarity and relevance. We hope these revisions align with your expectations. Below, you will find our detailed comments and responses to each of your reviews.
Comments 1: While the authors describe 'identifying risk factors, such as age, vascular malformations, and implantation technique' (line 61-62), the present SR-meta seems not investigated the vascular malformations, and implantation technique. Thus, the aim of current study may be not well supported through the Introduction section.
Response 1: Initially, vascular malformations and implantation techniques were not considered potential risk factors, due to a drafting error in the Introduction section. Furthermore, during the systematic review process, insufficient detailed information on vascular malformations was found in the included studies. Consequently, a statistical analysis of this specific factor was not possible.
On the other hand, we did investigate the role of implantation techniques, specifically comparing ERM and non-ERM techniques.
We will revise the manuscript to explicitly clarify these points, emphasizing that vascular malformations could not be assessed due to data limitations, and highlighting our analysis of implantation techniques (ERM vs. non-ERM).
Comments 2: The ethnicity or reported country may affect the prevalence of such adverse events as CVE.
Response 2: We fully agree with the reviewer that ethnicity or geographic origin could influence the prevalence of CVD in DBS surgery. Although we recorded country of origin in most studies, many of the included articles on patients with CVD lacked detailed data on ethnicity and uniformity in reporting demographic characteristics. This inconsistency prevented us from performing meaningful subgroup statistical analysis based on ethnicity or geography.
However, we noted that most studies originated from European and Asian centers. We acknowledge that this distribution likely reflects geographic patterns of research rather than deliberate selection bias.
In response to the reviewer's recommendation, we will explicitly highlight this limitation in the revised manuscript and suggest that future studies systematically report demographic data, including ethnicity, to allow for robust analyses of these critical factors.
Comments 3: If some of the included studies proposed countermeasures to prevent CVE, please consider to summarize them.
Response 3: Several studies briefly suggested general preventive strategies, such as meticulous blood pressure control during perioperative periods, careful management of anticoagulant and antiplatelet medications, utilization of advanced preoperative imaging for accurate trajectory planning, and precise electrode placement guided by MER.
However, these strategies were typically proposed narratively or theoretically, without robust statistical evidence or formal assessments within the included studies. Therefore, comprehensive quantitative evaluation of preventive measures was not feasible in our review.
Following the reviewer’s valuable suggestion, we will summarize these recommendations explicitly in the revised manuscript’s discussion section, emphasizing the necessity of prospective studies to systematically assess the effectiveness of these preventive measures in future research.
Round 2
Reviewer 3 Report
Comments and Suggestions for Authors
The 3rd comment from the previous version has not been adequately addressed. The author should remove any statistical comparisons and conclusions throughout the manuscript, including the abstract for the GPi vs STN, due to significant sample size mismatch. The rest of the comments have been addressed properly.
The analysis contrasts the CVE risks of STN and GPi targets; however, the number of GPi patients (n=170) is significantly smaller than that of STN (n=2,355), significantly reducing statistical power and making the GPi risk estimates extremely unstable (as evidenced by wide confidence intervals).
Author Response
Comment 1: The 3rd comment from the previous version has not been adequately addressed. The author should remove any statistical comparisons and conclusions throughout the manuscript, including the abstract for the GPi vs STN, due to significant sample size mismatch. The rest of the comments have been addressed properly.
The analysis contrasts the CVE risks of STN and GPi targets; however, the number of GPi patients (n=170) is significantly smaller than that of STN (n=2,355), significantly reducing statistical power and making the GPi risk estimates extremely unstable (as evidenced by wide confidence intervals).
Response 1:
Dear Reviewer,
Thank you very much for reviewing this comment.
Indeed, we have re-examined the manuscript and realized that in some sections, such as the Title, Abstract, Discussion, and Conclusion, we had included text referring to comparisons between STN and GPi (whether between the targets or the types of CVEs).
Therefore, we have modified these sections with the aim of removing such comparisons and presenting the results independently, specifically in the parts where the structures are mentioned.
Below, we present the modifications made:
Title
Risk of Cerebrovascular Events in Deep Brain Stimulation for Parkinson's Disease Focused on STN and GPi: Systematic Review and Meta-Analysis
Abstract
Lines 23-28: […] The overall CVE probability was 2.71% (95% CI: 2.27%–3.18%). Descriptive probabilities were 2.56% (95% CI: 1.94%–3.24%) for STN and 0.93% (95% CI: 0.00%–3.08%) for GPi. Hemorrhagic events were most common (STN: 2.47%; GPi: 1.98%), while ischemic events were rare (STN: 0.07%; GPi: 1.98%). Note that GPi estimates are based on a considerably smaller population and should be interpreted with caution. […]
Lines 30-34: […] Conclusions: Our results suggest that DBS in PD is associated with a relatively low CVE risk (~2.7%), with hemorrhage being the most frequent type; CVEs remain a potential risk factor. Comprehensive evaluation of patient-specific factors and further prospective studies focusing on CVE outcomes are essential to optimize DBS safety in managing PD.
Discussion
Lines 390-400: Current surgical techniques incorporating high-resolution imaging for trajectory planning and meticulous intraoperative execution help mitigate the inherent anatomical risks associated with DBS. The decision regarding which target to use for DBS is multifactorial. It is influenced by patient age, predominant symptoms (such as motor issues versus dyskinesia), medication-related goals, potential adverse effects, and the surgical team’s expertise [6,7,53,56]. For example, STN DBS is generally associated with reducing levodopa requirements, while GPi DBS may be chosen to address dyskinesia with a lower impact on cognitive or mood functions [40,55,57–59]. Our descriptive findings indicate that vascular safety is only one of several important factors in DBS surgery, underscoring the necessity of a tailored treatment approach. Although we report estimates for STN and GPi, it is important to note that the GPi data are derived from a much smaller patient sample; Consequently, these estimates should be interpreted cautiously.
Conclusion
Lines 513-515: Overall, these descriptive findings indicate that, although the overall probability of CVE in PD patients undergoing DBS is relatively low, CVEs must still be regarded as a potential risk factor during DBS procedures. […]